# Auditing the quality of epidemic decision-making in Somalia: a pilot evaluation

Abdihamid Warsame [1], Abdikadir Ore,[2] Abdullah Azad,[3] Farhan Hassan,[4] Karl Blanchet,[5] Jennifer Palmer,[6] Francesco Checchi[7]

For numbered affiliations see end of article.

**Correspondence to**
Dr Abdihamid Warsame; abdihamid.warsame@lshtm.ac.uk

## ABSTRACT

**Objective** To assess decision-making quality through piloting an audit tool among decision-makers responding to the COVID-19 epidemic in Somalia.

**Design and setting** We utilised a mixed-methods programme evaluation design comprising quantitative and qualitative methods. Decision-makers in Somalia piloted the audit tool generating a scorecard for decision-making in epidemic response. They also participated in key informant interviews discussing their experience with the audit process and results.

**Participants** A total of 18 decision-makers from two humanitarian agencies responding to COVID-19 in Somalia were recruited to pilot the audit tool.

**Outcome measures and analysis** We used thematic analysis to assess the feasibility and perceived utility of the audit tool by intended users (decision-makers). We also calculated Fleiss' Kappa to assess inter-rater agreement in the audit scorecard.

**Results** The audit highlighted areas of improvement in decision-making among both organisations including in the dimensions of accountability and transparency. Despite the audit occurring in a highly complex operating environment, decision-makers found the process to be feasible and of high utility. The flexibility of the audit approach allowed for organisations to adapt the audit to their needs. As a result, organisation reported a high level of acceptance of the findings.

**Conclusion** Strengthening decision-making processes is key to realising the objectives of epidemic response. This pilot evaluation contributes towards this goal by the testing what, to our knowledge, may be the first tool designed specifically to assess quality of decision-making processes in epidemic response. The tool has proven feasible and acceptable in assessing decision-making quality in an ongoing response and has potential applicability in assessing decision-making in broader humanitarian response.

## STRENGTHS AND LIMITATIONS OF THIS STUDY

⇒ The criteria for decision-making quality are listed and defined.
⇒ This study utilises mixed methods to both audit quality of decisions as well as shed light on the utility and feasibility of the tool.
⇒ This study is limited to auditing organisational decision-making and does not attempt to audit individual decision-makers.
⇒ Only decisions related to a single epidemic within one setting were audited.
⇒ The audit process was organisationally led in order to determine the feasibility of the audit.

## INTRODUCTION

Evaluating decision-making has been recognised as essential to improving health outcomes in a number of contexts.[1] In humanitarian and crisis contexts good decisions at programme, sector and response level are especially critical to saving lives and improving response.[2] Despite recent calls for concerted evaluation of decision-making in these settings,[3 4] the literature is still sparse. In crisis-affected settings experiencing epidemics, evaluations to date have primarily focused on establishing the extent to which epidemic response outcomes (eg, reduced transmission, improved case management) have been attained. Less attention has been given to evaluate the processes underlying these outcomes (eg, how response activities were decided and implemented).[5] Process evaluations have largely been conducted in high-income countries or after high-profile epidemics to retrospectively determine which decisions led to response failures.[6–8]

The need to improve decisions in epidemic settings is especially relevant considering the ongoing COVID-19 pandemic, in which decision-makers contend with a plethora of competing emergencies.[9] Decision support tools have been developed for a variety of settings and purposes but are particularly ubiquitous in corporate business management[10] and in the pharmaceutical and health technology sector,[11] for example in the development and licensing of medicinal products, equipment and diagnostics. They are less frequent in humanitarian contexts where they focus primarily on supporting decision-makers in optimising selected response

interventions through for example specifying target populations or modalities of delivery.[12] [13] To our knowledge there does not exist an evaluation tool that examines the quality of decision processes within epidemic settings. In a previous study in Somalia,we described factors relevant to COVID-19 decision-making processes; results suggested a need for such a tool.[9]

We thus developed a decision-making audit tool to support epidemic responders in assessing and improving the quality of organisational decision processes. Here, we report on a pilot application of this tool among two epidemic response organisations in Somalia, a country grappling with the ongoing COVID-19 pandemic and ongoing humanitarian crises.[14]

## AIM AND OBJECTIVES

The overall aim was to evaluate the utility and feasibility of a decision support tool for epidemic responders in humanitarian settings.

The specific objectives were to: (1) document the implementation of the tool among select epidemic responders in Somalia; (2) generate epidemic decision-making scores using the decision support tool; and (3) explore the feasibility and utility of the tool through key informant interviews of epidemic responders involved in the audit. We refer to objectives 1 and 2 as comprising the audit while objective 3 is referred to as the evaluation.

## METHODS
### Study design
This study used a mixed-methods programme/response evaluation design. It comprised quantitative and qualitative data collection to both assess the decision-making in epidemic response as well as the feasibility and perceived utility of the tool by intended users (decision-makers). We then revised the tool based on this data. The final structure of the decision-making evaluation tool was determined from input collected during the pilot evaluation.

### Patient or public involvement
Neither patients or the public were involved in the design of this study.

### Description of the tool
We developed an evaluation tool, protocol and Standard Operating Procedure (SOP) (online supplemental files 1–4) founded on previous reviews[15] and extensive fieldwork in an epidemic setting.[9] This foundational work resulted in findings of very low coverage of epidemic

**Table 1** Characteristics of 'critical' decisions

| Characteristic | Definition |
| --- | --- |
| Consequential | Decision shapes the response to a significant degree |
| Not reversible | Decision is difficult to overturn or reverse, at least in the short term |
| Strategic | Decision involves substantial shift in terms of action taken, resources committed or precedent set |
| Uncertain | Decision is made in context of substantial uncertainty with complex array of options |
| Reputationally risky | Decision entails a high level of organisational reputational risk |

evaluations globally, limited focus on decision processes, lack of standardised evaluation methods as well as an absence of a comprehensive evaluation framework suitable for epidemic response. As a result, we developed an Adaptive Epidemic Response evaluation framework[16] and drawing from an assessment of the COVID-19 response in Somalia,[9] we derived a decision-making framework. The tools and SOPs are based on this decision-making framework. The tool is comprised of three sections and is summarised below (figure 1).

The first section (Part A) of the tool is the context analysis, which requires information on the historical, geographic and health context in which the epidemic is occurring.

The second section (Part B) entails identification of critical decisions in the response to the specific epidemic. These critical decisions are identified and selected with reference to five Critical Decision Characteristics that differentiate critical decisions from minor or low impact decisions (table 1).[2]

In the third section of the tool (Part C), users assess each of the selected decisions against the criteria for 'quality' decision-making. Although there is no agreed definition of a good quality decision,[17] we have derived a number of defining criteria from previous published research on health prioritisation,[18] organisational decision-making[17 19 20] as well as decision-making in emergencies.[21] The 11 criteria are grouped into four dimensions: transparency, contestability, accountability and rigour (table 2).

Users are required to rate on a Likert scale (a type of linear rating scale commonly used to measure respondents' opinions or attitudes), the strength of evidence supporting the fulfilment of each criterion. They are also expected to provide reference to documentary or

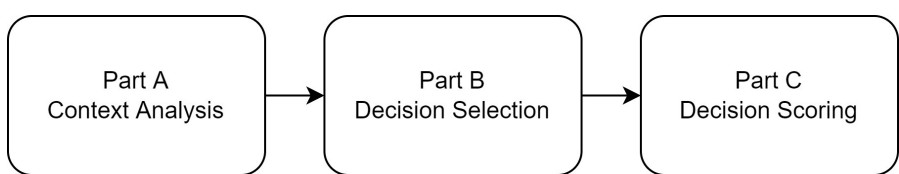

**Figure 1** Structure of the audit tool.

**Table 2** Criteria for assessing decision quality

| Dimension | Criteria | Description |
|---|---|---|
| Transparency | Inclusivity | The extent to which the process was inclusive, reflected in heterogeneity in rank and roles among decision-makers involved. |
| | Use of explicit decision-making criteria | The extent to which the goals and objectives of the decision were clearly prespecified. The absence of post-decision rationalisation. |
| | Following clear process or method | The extent to which a priority setting process was in place, reflected in demonstrated use of priority setting frameworks, decision trees or other mechanism. |
| | Use of mechanism to publicise rationale | The extent to which clear documentation on the decision exists as well as the method used to communicate decisions. |
| Contestability | Opportunity for revision | The extent to which there existed scope to revise and overturn a decision including the debating of alternatives and description of how consensus was reached. |
| | Was the decision devolved? | The degree to which participants in closest proximity to the epidemic (eg, subnational level) or local technical experts participate in the decision, including consideration of rank. |
| Accountability | Engagement with affected communities | The degree to which affected communities were involved in the response decision-making including at a minimum whether they were informed of the response activities and what effect this notification had on the communities. |
| Rigour | Explicit outcome | The extent to which intended outcomes of the decision were clearly articulated, including through setting of targets. |
| | Feasible outcome | The extent to which feasibility was considered in decision-making including debating of alternatives. |
| | Strengthens healthcare system | The extent to which the decision was in-line with wider strategy including the strengthening of the health system |
| | Evidence based | The extent to which the decision was based on strong public health rationale and robust scientific information. |

observational evidence to support their rating. Users' individual scores are then aggregated, and a summary score is generated.

### Data collection
#### Audit

We invited three organisations actively engaged in the COVID-19 response in Somalia to pilot the audit tool. After separate presentations in which the protocol and study objectives were explained by the first author, we partnered with two organisations (WHO Somalia and CARE Somalia) in September 2021. Both organisations then nominated audit focal persons tasked with recruiting relevant colleagues (a 'decision-making committee'), gathering the necessary documentation and completing Part A of the tool. Focal persons were instructed to recruit colleagues who had an active role in decision-making

within the response as well as to compile key documents which informed or documented decisions. The location and modality of the audit (combination of face to face and remote sessions) and timeline were jointly determined by the audit focal persons and the first author. Study information and consent forms were shared, and written consent was obtained from all participants in the pilot.

The decision-making audit and feasibility evaluation took place in Garowe, Somalia from 8 to 22 November 2021 with CARE Somalia and in Mogadishu from 22 November to 8 December with WHO Somalia. The audit was led by each organisation's focal persons with the facilitation of the first author. Each audit was comprised of three group sessions with the decision-making committee interspersed with individual sessions (figure 2). The first

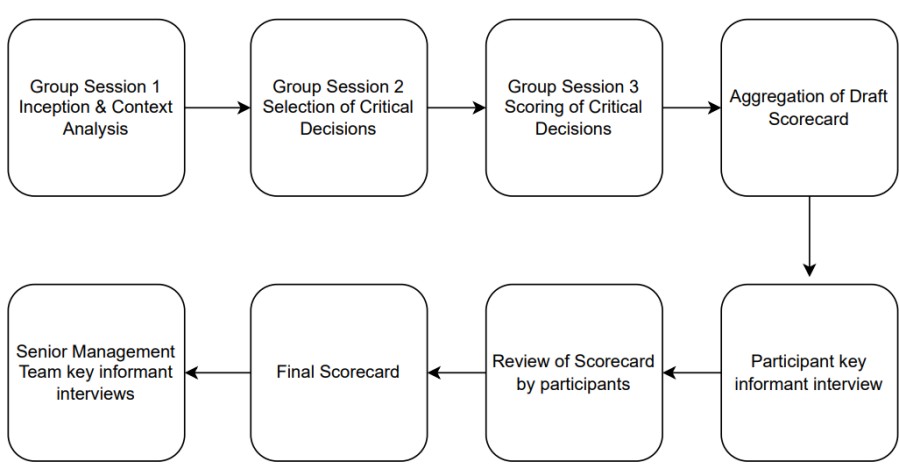

**Figure 2** Data collection timeline of evaluation.

group session introduced the audit approach, tool, timeline and expected outputs. Participants were then asked to review Part A (context analysis) of the tool and incorporate any changes before the next group session. In the second group session, participants were introduced to Part B (selection of critical decisions) and were tasked with individually generating their perceived list of critical decisions. In the third group session, participants were asked to each present and advocate for their selection of critical decisions to the wider group. The group was then tasked with forming a consensus on at least three of the decisions to carry forward to the last stage. After consensus was reached, participants were asked to complete Part C of the tool for each decision in which they assessed the availability and strength of evidence supporting the fulfilment of various quality decision-making criteria. Audit focal persons and the London School of Hygiene and Tropical Medicine (LSHTM) researcher then aggregated the scores and created a draft scorecard.

### Evaluation

All audit participants were then invited to take part in a key informant interview to expound on their selected decisions and explore their views on the audit process and tools. After the key informant interviews, the draft scorecard was shared with all participants to provide individual written feedback on the results. The final scorecard was then developed by the audit focal persons and first author and shared with the senior management of the respective organisation. Finally, key informant interviews were conducted by the first author with senior managers who were not involved in the other aspects of the audit and evaluation to solicit their views on the results and their understanding of the process. The tool and related documents were then updated based on pilot findings.

A total of 18 key informant semistructured interviews were conducted in English or Somali by the first author either face-to-face or via Zoom (table 3). The first author is a fluent Somali speaker who has substantial experience in health evaluations and epidemic response in the Somali context. The topic guide covered participants' views on the feasibility of the audit process including time and human resources required, the utility of the audit tool in

**Table 3** Data collection by method and type of participant or setting

| Primary data collection | Type of participants | Number of participants |
| --- | --- | --- |
| Key informant interviews | WHO audit team | 7 |
| | WHO Staff—other | 2 |
| | WHO senior management | 2 |
| | CARE Somalia audit team | 5 |
| | CARE Somalia senior management | 2 |
| Group sessions | WHO audit team | 7 |
| | CARE audit team | 6 |

understanding and improving decision-making as well as recommendations for future iterations.

Interviews were recorded for transcription and analysis purposes. Each interview took approximately 30–45 min and participants were given the option of complete or partial anonymity in which they considered whether they would like their name or only their role to be published. A total of 13 participants took part in the group sessions and notes were taken by the first author. All participants in the CARE audit were national staff members while the majority of WHO participants were international (85%).

### Data analysis
#### Generation of audit tool scoring

The scorecard for the audit was produced by calculating for each of the 11 criteria assessed, the average of the individual participants scores. Average scores were also presented according to the four dimensions listed in table 2.

#### Evaluation of the audit implementation

In order to evaluate the implementation of the audit tool as it relates to feasibility and utility, interview transcripts and group session notes were analysed using a thematic approach[22] using Nvivo software. Emerging themes were grouped into two categories: those relating to identified critical COVID-19 decisions and those relating to the implementation of the audit. Lastly the validity of the tool was evaluated. The inter-rater agreement of each organisation's scoring was calculated using Fleiss' Kappa.

In the following Results section, we focus on the implementation of the audit while details of specific decisions can be found within the decision-making scorecards in the supplemental material (online supplemental files 5 and 6).

## RESULTS
### Implementation of the decision-making audit

Participants in both organisations opted for a mixture of face-to-face and remote (video conference) data collection citing the busy schedule and geographic dispersion of the participants. Although preparatory material including documentation was shared prior to the arrival of the LSHTM focal person, some participants were not immediately aware of the purpose of the audit. The delayed setup of shared folders containing key audit documentation proved difficult for participants' engagement with the audit and subsequent evaluation. Furthermore, the period in which this audit took place coincided with multiple projects including end-of-year evaluations, annual planning and multiple prescheduled activities, hindering the timely recruitment of participants. Many participants were not able to be physically present in the location of the audit due to competing engagements but were nevertheless present through remote means.

### Decision selection

CARE participants had difficulty identifying critical decisions with individual participants submitting on average

**Table 4** List of selected critical decisions by organisation

| CARE Somalia | WHO Somalia |
| --- | --- |
| Closure of offices and restriction of staff movement | Scale up of case management through improving access to therapeutic oxygen |
| Modification of nutrition programme guidelines to be COVID-19 sensitive including changing patient assessment, facility management and outreach procedures | Establishment of three key PCR labs in Mogadishu, Garowe and Hargeisa to strengthen diagnostic and surveillance capacity |
| Scale up of COVID-19 response activities such as community outreach, contact tracing, provisioning of Infection Prevention and Control (IPC) supplies in Sool and Sanaag regions | Implementation of the Incident Management Support Team to coordinate COVID-19 response |
| | Establishment of rapid response teams to scale up surveillance capacity within high priority districts |

only one critical decision for consideration. However, consensus was reached fairly quickly once the proposed decisions were deliberated by the wider group. In contrast, WHO audit participants put forward 15 decisions for consideration. Consensus took longer to achieve as participants vigorously advocated for their proposed individual decisions. Eventually the group reached consensus on four decisions to proceed to the next stage (table 4). In both groups, moderate reference was made to the Critical Decision Characteristics (table 1), with participants arguing for decisions in their particular area of work. Participants in WHO were particularly inclined to view decisions in terms of their alignment with the Incident Management Support Team (IMST) Pillars.[23] Additionally, there was an effort by some participants to reach consensus by collapsing together multiple decisions into a single decision in order to capture all opinions. In both WHO and CARE evaluations, participants focused on positive decisions (ie, a decision to take an action) and

may have overlooked negative decisions (ie, decisions not to take action).

### Decision scoring

Participants in the CARE audit on average scored their decision-making quality lower than those of WHO. Both organisations scored lowest in accountability to target populations. The scoring of WHO participants on the decision to focus on case management through oxygen scale up demonstrates the low scoring of accountability relative to the other dimensions of quality decision-making (figure 3). Full details can be found in the supplementary materials (online supplemental files 5 and 6).

Participants in CARE scored their decisions lower in contestability compared with WHO. Both organisations rated themselves highly in transparency, particularly in the inclusivity criteria. However, the key stakeholder mentioned under these criteria differed, with CARE highlighting the need to improve inclusive decision-making within the organisation (national staff vs country office senior management), while WHO referred to inclusivity with government authorities. Lastly, CARE participants rated their decision-making as less rigorous than WHO's, citing their unfamiliarity with the evidence underlying the decisions. By contrast, WHO participants cited use of the latest scientific findings and alignment with health systems strengthening frameworks.

### Inter-rater agreement

There was very low inter-rater agreement among participants in both WHO (Kappa=−0.000948, p=0.977) and CARE (Kappa=0.0684, p=0.172) audit groups with a slightly higher inter-rater agreement among CARE participants, indicating high heterogeneity of scores.

### Perceptions of the audit tool
#### Utility

Participants described clearly seeing the purpose and value of the audit. For example, one participant from WHO said, '*We learned that decisions are not just simply*

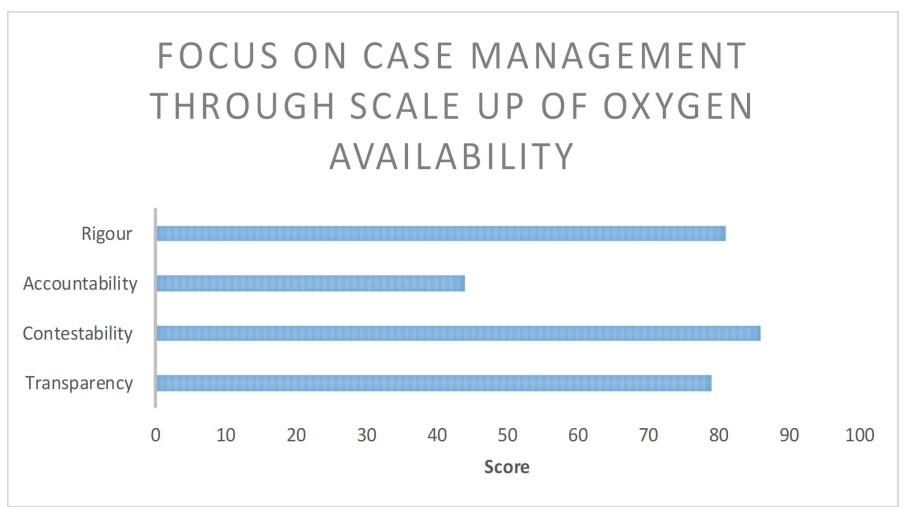

**Figure 3** Extract of decision scorecards where accountability was rated low.

*taken as decisions, but you have to have a clear decision-making process, a methodology, or criteria, or something, but not necessarily that we just come and say, Oh, let's do this and that'.* Participants articulated the ongoing need for such an exercise, arguing: '*If there's no understanding of how a decision happened, there's no way to improve on it*' (WHO Participant 5). Critically, they understood that this exercise was meant as an internal audit: '*It was really for us to be able to critically evaluate ourselves as an organisation*' (CARE Participant 3). Some participants also saw the potential utility in a wider range of contexts, beyond epidemics: '*I think (the audit tool) can apply to any decision in humanitarian emergency response*' (WHO Participant 2).

Participants mentioned learning new things about the decision process including recognising neglected characteristics of good decision-making:

When I was trying to fill in some of the scores, I realised that these are very important (aspects) that were neglected before. For example, devolving certain decisions, it is very important. Not just like you sit up at national level, you decide for anyone and you decide for people at all levels, I think if we devolve certain functions it is good. WHO Participant 1

They also were able to identify areas for improvement. '*I think for accountability most of the evidence was very minimal. It just got me thinking of ways in which we can involve the communities in decision-making*' (WHO Participant 4).

Participants also highlighted that the audit was useful in allowing them to be more reflective. '*It helped me look at things objectively. If I was not part of the Incident Management Support Team (IMST) would I have advocated for this decision or not?*' (WHO Participant 4). They highlighted the relevance of the decision-making critical characteristics as useful in their work and recognised these as reflective of best-practice: '*I think the tool helped a lot to give me a reflection as to what should be done. I think I learnt a lot going through the questions*' (WHO Participant 1).

Participants mentioned the gradual understanding they accrued throughout the audit process as strengthening their engagement with the audit. '*I think some of it was clear, but some of it was also not very clear. I think I understood the methodology with time, of course, when you explained and when we were having the group discussions. That is why in the last forms, I revised and added a few components and sent back to you. I think as usual when people read something, it might not be clear in the first place, but when you explain it, it becomes clear*' (WHO Participant 4).

Participants also noted the benefit of the group format in deriving new insights and creating a useful forum for reflection to review decisions that otherwise did not exist in their typical organisational practices:

We all had our own ideas, but when we met as a group, there were a few other things that came out that probably I didn't think about as an individual, which were very valid. WHO Participant 4

I also liked that we were able to meet as a group and look at these things because to be very honest, I don't feel like we have that as a group, as an IMST team to review the decisions that we made or look at what we could have done differently, or whether we felt this decision was important or not. This was a very unique exercise and I'm glad that we got an opportunity to do this. WHO Participant 7

Participants expressed their commitment to use the lessons learnt in future decision-making:

I will inform my colleagues, 'Please, let us have certain criteria to use, let us make our methodology clear, and let us document everything,' because at some point down the road, we should know our decision-making process. This has helped a lot and will change a lot in the future. WHO Participant 5

However, participants mentioned that there was a certain element of subjectivity which was difficult to overcome:

It might be very difficult to understand exactly the reality on the ground, because it depends on the person. Not only the person but also, the involvement of the person in the response activity, and also the understanding of the context. If you are someone who is quite new to Somalia, and maybe who doesn't have detailed involvement, you might give a different ranking than a person who's spent more time in Somalia and directly engaged in response activity. WHO Participant 2

### Feasibility

In terms of the audits' feasibility, participants expressed that they were easily able to comprehend the process. '*[The tool] was very understandable and also, it was very user friendly*' (CARE Participant 1). However, a few participants noted some areas for improvement: '*I'm looking at this tool being used for an outbreak. If that is to happen, as I said, we need to really condense some aspects, and lump some of these criteria together to cut on time*' (CARE Participant 3).

In terms of time, participants expressed that the duration of the audit was acceptable but that the time of year chosen for the implementation could have been improved. '*In terms of length of the time, this is fine. This amount of time we need to give. The issue is the timing of the programming, because in some of the months, we are very busy, like last quarter of the year we are very busy*' (CARE Participant 5).

As a result, some participants expressed their confidence that they could implement this audit independently at a more opportune time. However, others expressed their preference for this work to be led by an independent evaluator.

As the independent outsider, everybody can open up to you, and this is the best advantage I can see. I also saw something similar in my previous organization,

that again, it is the power dynamic or the personality thing that not everybody speaks openly, or the culture doesn't allow it but when somebody from outside comes, everybody opens up. At the end, you come up with some really good findings and recommendations. WHO Participant 1

In their engagement with the audit tool, participants faced a challenge in coming to a consensus on the limited number of decisions to score. *'Considering the response that we are doing in the country, it might be a bit challenging to come up with one or two out of so many critical decisions'* (WHO Participant 2).

Additionally, there were challenges in accessing documentation relevant to the audit. *'Maybe the big challenge is getting the evidence, documentation, preparing the documentation, extracting the email, finding the document, it might be time-consuming'* (WHO Participant 3). Where documents did exist, some participants stated that not all staff may have access. Lastly, some participants highlighted language or conceptual understanding barriers as challenges in the audit. *'At some point, I felt like we were not looking at the decisions, but we were looking at pillars'* (WHO Participant 4).

### Selection of decision-making committee
Participants noted that the composition of the decision-making committee had the potential to affect the audit outcome because of a lack of detailed knowledge on how decisions were made. *'If somebody is not involved in a certain pillar, they would struggle to look for evidence or they might not be able to direct somebody to a particular document'* (WHO Participant 2).

They mentioned that all who were part of the audit had some role to play in COVID-19 decision-making but conversely not all who had decision-making roles participated in the audit.

Some participants highlighted the potential of overlooking critical decisions when the primary responsible staff member was not present in the audit.

Later I thought about COVID-19 vaccine, and I wondered if we could have included the focal person and whether they would have felt that the introduction of the COVID-19 vaccine into the country was a critical decision. I thought about it, and I thought maybe nobody mentioned it, or nobody talked much about it during the discussion because the focal point was not there. WHO Participant 4

### Participant recommendations
Participants made several recommendations to improve the utility and feasibility of the audit for future iterations. These included broadening the scope of participants by more clearly outlining who should participate.

Participants also suggested incorporating the emergency response cycle into the evaluation tool. *'There are emergency response cycle phases such as preparedness, implementation, monitoring, and evaluation […] There are critical decisions at each phase which need to be evaluated'* (WHO Participant 2)

They also made several suggestions to improve the implementation including ensuring more face-to-face rather than virtual interactions among participants and undertaking a joint review of documentation to improve participant familiarity with the available documents.

### Views of senior management
Senior management largely agreed with the findings of their respective scorecards. They did, however, provide further detail and rationale for some aspects of the selected decisions. For example, senior managers in both organisations cited mitigating factors in explaining lower-than-expected scoring in some dimensions. Referring to closure of offices and restriction of staff movements, one CARE senior manager explained:

When the first case of COVID-19 was reported in Somalia in March, I think it was around 16, 17. There was a bit of panic in that it was not business as usual. When it comes to between life and death, certain positions have to be taken. As a result of that, yes, the senior management team made a decision to ensure that because we are accountable to everyone and every staff, and every staff exposed to any risk as a result of negligence of the organization, then the organization takes responsibility. CARE Senior Manager 1

A WHO senior manager explained the circumstances behind a decision to increase testing capacity:

Obviously there's scope for improvement, but sometimes when we are in a pressured situations and we see that the PCR based labs are overwhelmed and at the peripheral level there is no access or testing facilities are not there, and the GeneXpert machines are also not functioning, then sometimes we make decisions and implement swiftly, and then [later] we provide other evidence and information of the value and effectiveness of those strategies. WHO Senior Manager 1

They did however agree on gaps highlighted through the scorecard stating: *'I would say that [decision-making] can become more inclusive. I think that we are just talking internally amongst ourselves and we are not engaging sufficiently with our government counterparts. I think that that's a fundamental mistake that we are doing, and we continuously do so, because of maybe convenience, because of maybe comfort'* (WHO Senior Manager 2).

### DISCUSSION
This pilot evaluation of a tool to audit decisions taken during epidemic response demonstrates that the tool can be successfully deployed even in the midst of an ongoing response and even in circumstances where responders are dealing with unusually high workload. The audit also elicited very positive user feedback with

participants expressing a willingness to implement it in future epidemic responses. Importantly, participants viewed the audit as reflecting good decision-making practice indicating high levels of acceptability.

As populations affected by epidemics continue to grow,[24] there is an urgent need to improve epidemic response particularly through improved decision-making. Much of the focus has been on improving decision outcomes and less attention has been paid to decision-making processes.[9] However, strengthening decision processes can improve decision-making quality[25] and, accordingly, outcomes[26] including more efficient allocation of resources, improved accountability and greater coordination.

The literature on decision-making in emergencies has been underpinned by an analytically oriented[27] conceptualisation of decision-making in which the 'correct' decision is sought.[28 29] This is in contrast to the process conceptualisation in which the decision is pursued correctly. Towards this end, decision support approaches have been developed to optimise the decision-making process to achieve efficient[2] and timely decisions,[30] but have largely not considered the aspect of quality. The evaluation of this tool within the COVID-19 response in Somalia contributes towards filling this gap.

The audit sheds light on shortcomings in the quality of decision processes within piloting organisations and provided participants opportunity for reflection and key areas for improvement. The tool was found to be highly adaptable as it allowed organisations to evaluate decisions that were considered significant by their staff. Nevertheless, while the audit instructed organisation to select participants who were largely reflective of staff making or implementing response decisions, some participants expressed reservations about the final composition of the decision-making committee. The scoring of some decision dimensions was however largely reflective of the committee's experiences and characteristics. For example, among CARE participants, the lower scoring on the contestability of some decisions might be due to their roles as national staff who may not have had sufficient opportunity to contest decisions. The rollout of the tool allowed for gradual understanding to develop and for participants to question their basic assumptions. For example, participants were able to acknowledge elements that were absent from their decision-making process. They were also surprised by the paucity of documentary evidence within their organisations and were able to contrast this with their initial self-assessed high scoring. The lower-than-expected agreement among raters could be possibly due to the lack of consensus on what various levels of evidence represented. While limited documentation does not necessarily equate with poor decision processes, it does make subsequent evaluations more challenging.

Furthermore, the audit tool was found to be highly flexible as users highlighted its potential utility in assessing the quality of decision-making in broader humanitarian response by generating a quantitative measure of decision quality that can allow for tracking over time. In addition to the retrospective assessment of decisions, the tool can also be used in real time to improve decision processes.

## Limitations

Although this pilot was undertaken within two organisations, it assessed decision-making related to a single epidemic within one country. While this pilot study was facilitated by the first author who was external to both organisations, future iterations led entirely by internal staff members may be more vulnerable to censorship if results reveal low decision quality. However this is a challenge that is present in the global health arena in general[31] and a number of resources have been published to strengthen independence of evaluations.[32–34] Additionally, much of the scoring relied on the subjective assessment of the individual rater and may thus have been reflected in the lower-than-expected inter-rater correlation. Furthermore, this pilot focused on face validity (the extent to which a tool appears to measure a concept) rather than on content validity which requires further exploration and methods. Additionally, we did not use quantitative methods such as factor analysis to reduce the number of items within the critical decision criteria as there were too few items. Instead, we relied exclusively on direct feedback from the participant interviews. Definitions for the decision-making criteria were not exhaustive and could have been further expanded. For example with regards to community engagement criteria, we provided a minimum definition rather than broader definition.[35] Finally, the methods described in this study assessed only organisational decision-making rather than individual and as such cannot be used to assess the decision-making of individual epidemic responders.

## Recommendations

The audit can be further piloted in a wider range of crisis settings and among different response actors to ascertain its feasibility and utility in diverse settings. Additionally, the audit should be conducted internally by response actors in order to compare how the audit is implemented when it is entirely independent of an external facilitator. Furthermore, the audit should be conducted periodically in order to determine whether there has been a quantitative change in the decision quality scoring. Lastly, the audit tool could also be integrated within the WHO recommended intra-action reviews[36] as well as after action reviews[37] for health emergencies.

## Conclusion

Strengthening decision-making processes is key to realising the objectives of epidemic response. This pilot evaluation contributes towards this goal by the testing what, to our knowledge, may be the first tool designed specifically to assess quality of decision-making processes in epidemic response. The tool has proven feasible and acceptable in assessing decision-making quality in an ongoing response

and has potential applicability in assessing decision-making in broader humanitarian response.

## Author affiliations

[1] Department of Infectious Disease Epidemiology, London School of Hygiene and Tropical Medicine, London, UK
[2] Humanitarian Department, Care International Somalia, Garowe, Somalia
[3] Health Emergencies Department, World Health Organization Somalia, Mogadishu, Somalia
[4] Health Systems Strengthening Department, World Health Organization Somalia, Mogadishu, Somalia
[5] Faculty of Medicine, Geneva Centre of Humanitarian Studies, University of Geneva, Geneve, Switzerland
[6] Department of Global Health and Development, London School of Hygiene and Tropical Medicine, London, UK
[7] Faculty of Epidemiology and Public Health, London School of Hygiene and Tropical Medicine, London, UK

**Acknowledgements** We would like to acknowledge Md Mamunur Malik WHO country representative and Iman Abdullahi CARE country representative for agreeing to pilot this tool in their respective organisations. We would like to thank participants for sharing their insights.

**Contributors** This study was designed by AW with input from FC. The data collection was undertaken by AW, AO, FH and AA. The analysis was undertaken by AW with input from KB, FC and JP. The first draft was written by AW. All authors have contributed significantly to the final draft. AW is the guarantor of this study.

**Funding** This work was supported by UK Research and Innovation as part of the Global Challenges Research Fund, grant number ES/P010873/.

**Competing interests** None declared.

**Patient and public involvement** Patients and/or the public were not involved in the design, or conduct, or reporting or dissemination plans of this research.

**Patient consent for publication** Not applicable.

**Ethics approval** Ethical approval for this study was obtained from the National Institute of Health Research of the Federal Republic of Somalia (Ref: NIHS0102208) and the ethics review committee of the London School of Hygiene & Tropical Medicine (Ref: 26369). Informed written consent was taken from all key informants.

**Provenance and peer review** Not commissioned; externally peer reviewed.

**Data availability statement** All data relevant to the study are included in the article or uploaded as supplementary information.

**ORCID iD**
Abdihamid Warsame http://orcid.org/0000-0003-2524-256X

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
