## [Reviewer comments · BMJ Open]

ARTICLE DETAILS

TITLE (PROVISIONAL)	Auditing the quality of epidemic decision-making in Somalia: a pilot evaluation
AUTHORS	Warsame, Abdihamid; Ore, Abdulkadir; Azad, Abdullah; Hassan, Farhan; Blanchet, Karl; Palmer, Jennifer; Checchi, Francesco

VERSION 1 – REVIEW

REVIEWER	Eric Christensen University of Minnesota , Health Services Management, College of Continuing and Professional Studies
REVIEW RETURNED	21-Jul-2022

GENERAL COMMENTS	This paper piloted an audit tool to assess decision making quality during an epidemic. I'm not a qualitative researcher, so I'm probably the wrong person to review this manuscript. Nevertheless, I have reviewed, and I have a few comments that I hope will be helpful. First, decision-making quality is very subjective. The authors should be more explicit about what this is and is not. The authors note that quality of organizational decision processes. Again, what does that mean and how do you judge it good or bad? Second, the authors noted that some KIIs were conducted in English and some in Somali. Were there any systematic differences by language? How many were done in each language? Third, all 18 had KIIs, but only 13 had group sessions. were there differences between those who participated in group sessions and those who did not? Fourth, there seemed to be differences between the CARE and WHO groups. As there were only 2 groups, it seems difficult to draw a lot of conclusions given the differences. I don't know what to make of the differences between the CARE and WHO teams. Finally, I question how much any organization will get away from reacting to an epidemic to employ the audit tool. Some discussion of this difficulty would be useful.
--

REVIEWER	Landry Mayigane World Health Organization, Country health emergency preparedness and IHR
REVIEW RETURNED	10-Oct-2022

GENERAL COMMENTS	The manuscripts requires major revisions in terms of structure, definition of terminologies (Evaluation vs Audit), methodology,
---

	presentation of results and discussion. The reviewer provided a marked copy with additional comments. Please contact the publisher for full details.
--	--

VERSION 1 – AUTHOR RESPONSE

Comment	Response
First, decision-making quality is very subjective. The authors should be more explicit about what this is and is not.	We have provided a definition of decision-making quality on page 5 lines 130-134 including appropriate references.
The authors note that quality of organizational decision processes . Again, what does that mean and how do you judge it good or bad?	Thank you for your comment. We have provided this explanation in the manuscript (page 5 lines 130-134). Briefly: this paper is predicated on a previous paper in which we describe decision-making for COVID-19 response in Somalia. In the course of this paper, we have developed a decision-making framework which outlines the different dimensions and factors affecting decision-making. From these results as well as from a review of the literature, we have developed our decision-making audit tool in which criteria for good decision-making processes are listed. We judge a good decision-making process in the course of the audit as one in which there is documentary evidence for the fulfilment of the criteria.
Second, the authors noted that some KIIs were conducted in English and some in Somali. Were there any systematic differences by language? How many were done in each language?	Overall, 5 KII were conducted in Somali and 13 were done in English. The KII conducted in Somali were of national or local staff members while those done in English were a mixture of local and international staff.

Third, all 18 had KIIs, but only 13 had group sessions. were there differences between those who participated in group sessions and those who did not?	The group discussions excluded senior management and were comprised of 'rank and file' staff. This was done in order to allow for more open discussion.
Fourth, there seemed to be differences between the CARE and WHO groups. As there were only 2 groups, it seems difficult to draw a lot of conclusions given the differences. I don't know what to make of the differences between the CARE and WHO teams.	Thank you for your comment. The WHO and CARE groups were chosen for their similarities as well as differences. They were both international organisations responding to COVID-19 in Somalia with a history of humanitarian response. However, there are key differences between the teams: CARE is a nongovernmental organisation operating at a regional level, whilst WHO is a UN organisation operating at a national level. The CARE teams comprised a larger proportion of local staff compared to WHO. The purpose of the deliberate variation between the groups was to evaluate the audit tool in different contexts. It was not to directly compare the two organisations' decision-making processes. We have made recommendations to further ascertain the audit tool's utility and feasibility through use in more diverse settings including with different organisations, as well as in different countries in response to different emergencies.
Finally, I question how much any organization will get away from reacting to an epidemic to employ the audit tool. Some discussion of this	Thank you for highlighting the importance of feasibility. Ascertaining the feasibility of the audit tool implementation was one of our primary objectives in this study. We have made sure to elicit participants feedback and provided this information in the results section, subsection- feasibility. Participants provided their views on human resources required, time commitment amongst other factors. Participants noted that it was possible to use this audit independently of an external facilitator. In terms of time commitment, participants made suggestions for improvement including condensing some sections of the tool and deploying the tool during a time period that does not conflict with other evaluations.

difficulty would be useful.	
The manuscripts requires major revisions in terms of structure, definition of terminology (Evaluation vs Audit), methodology, presentation of results and discussion. See the attached file for details	We note with thanks your highlighting of the usage of the terms evaluation and audit. We use the term audit to refer to the assessment of decisions against the criteria set out in the tool. We use the term evaluation to refer to the assessment of the process of implementing the audit tool (the audit) with regards to the feasibility and utility as perceived by participants. The term pilot evaluation is used in the title to reference that this study is the first instance in which the audit implementation is being assessed. In the methodology section, we have addressed your comment related to the composition of the "panel". The panel is comprised of audit participants. Earlier in the manuscript we referred to this group as the "decision-making committee". We have decided to henceforth use only the term "decision-making committee" in order to maintain consistency. We have added this explanation in the aims and objectives section on page 4 of the manuscript. We have not found any details of major revisions related to the presentation of the results, methodology and discussion within the attached file.
I am not sure we need this document published as an annex here. It should be added to the list of references with proper citation and link where to retrieve it.	We note with thanks your suggestion to remove the Audit SOP from the supplementary material. However, we feel it would be better to publish this SOP together with the manuscript and audit reports rather than separately in order to ease dissemination.